# Lichen or Associated Micro-Organism Compounds Are Active against Human Coronaviruses

**DOI:** 10.3390/v15091859

**Published:** 2023-08-31

**Authors:** Lowiese Desmarets, Marion Millot, Marylène Chollet-Krugler, Joël Boustie, Charline Camuzet, Nathan François, Yves Rouillé, Sandrine Belouzard, Sophie Tomasi, Lengo Mambu, Karin Séron

**Affiliations:** 1Univ. Lille, CNRS, Inserm, CHU Lille, Institut Pasteur de Lille, U1019—UMR9017—Center for Infection and Immunity of Lille (CIIL), F-59000 Lille, France; lowiese.desmarets@ibl.cnrs.fr (L.D.); yves.rouille@ibl.cnrs.fr (Y.R.); sandrine.belouzard@ibl.cnrs.fr (S.B.); 2Univ. Limoges, Laboratoire LABCiS, UR 22722, F-87000 Limoges, France; marion.millot@unilim.fr (M.M.); lengo.mambu@unilim.fr (L.M.); 3Univ. Rennes, CNRS, ISCR (Institut des Sciences Chimiques de Rennes)—UMR 6226, F-35700 Rennes, France; marylene.chollet@univ-rennes.fr (M.C.-K.); joel.boustie@univ-rennes.fr (J.B.); sophie.tomasi@univ-rennes.fr (S.T.)

**Keywords:** antiviral, natural product, coronavirus, HCoV-229E, SARS-CoV-2, lichen compounds

## Abstract

(1) Background: Since the emergence of SARS-CoV-2, responsible for the COVID-19 pandemic, efforts have been made to identify antiviral compounds against human coronaviruses. With the aim of increasing the diversity of molecule scaffolds, 42 natural compounds, of which 28 were isolated from lichens and 14 from their associated microorganisms (bacteria and fungi), were screened against human coronavirus HCoV-229E. (2) Methods: Antiviral assays were performed using HCoV-229E in Huh-7 and Huh-7/TMPRSS2 cells and SARS-CoV-2 in a Vero-81-derived clone with a GFP reporter probe. (3) Results: Four lichen compounds, including chloroatranol, emodin, perlatolic acid and vulpinic acid, displayed high activities against HCoV-229E (IC_50_ = 68.86, 59.25, 16.42 and 14.58 μM, respectively) and no toxicity at active concentrations. Kinetics studies were performed to determine their mode of action. The four compounds were active when added at the replication step. Due to their significant activity, they were further tested on SARS-CoV-2. Perlatolic acid was shown to be active against SARS-CoV-2. (4) Conclusions: Taken together, these results show that lichens are a source of interesting antiviral agents against human coronaviruses. Moreover, perlatolic acid might be further studied for its pan-coronavirus antiviral activity.

## 1. Introduction

The ongoing pandemic of COVID-19 caused by severe acute respiratory syndrome coronavirus 2 (SARS-CoV-2) has highlighted the lack of effective antiviral drugs against human coronaviruses. Coronaviruses are members of the *Coronaviridae* family, divided in four genera: Alpha-, Beta-, Delta- and Gammacoronavirus. Seven Alpha- and Betacoronaviruses have been identified to infect humans. Four of them are associated with mild common-cold-like disease (HCoV-OC-43, HCoV-229E, HKU-1, HCoV-NL-63). The other three, including SARS-CoV, Middle East respiratory syndrome coronavirus (MERS-CoV) and SARS-CoV-2 have been responsible for outbreaks of severe pneumonia-like illness. To date, MERS-CoV and SARS-CoV-2 are still circulating. Coronaviruses are viruses with zoonotic origin, and all of them have a wild animal ancestor [1]. They are enveloped positive single-stranded RNA viruses. Their genome of approximately 30 Kb encodes for structural, non-structural (Nsp) and accessory proteins [2]. The structural proteins, membrane (M), envelope (E) and spike (S), are present at the surface of the virion, and the nucleoprotein N is associated with the RNA genome in the particle. The 16 Nsp are derived from two polyproteins, pp1a and pp1ab, which are cleaved by viral proteases and are involved in the formation of replication–transcription complexes [3]. Accessory proteins are involved in immune response escape and other not yet fully explored functions.

The coronavirus’ life cycle can be roughly divided in three main steps: entry, replication/translation and assembly/secretion [2]. The S protein plays a major role in entry because it interacts with cellular receptors and mediates the fusion of the virus with the cellular membranes. Two entry pathways have been described for HCoVs: direct fusion at the plasma membrane mediated by the cellular protease TMPRSS2, or endocytosis and fusion with the endosomal membrane mediated by cathepsins [4]. Cellular receptor usage is virus-dependent, being the angiotensin converting enzyme 2 (ACE2) for SARS-CoV and SARS-CoV-2, the dipeptidylpeptidase IV (DDP4) for MERS-CoV and the aminopeptidase N (APN) for HCoV-229E. Once the viral genome is released in the cytoplasm, pp1a and pp1ab are translated, and the resulting Nsp forms the replication–transcription complex (RTC), enabling the replication of the viral RNA genome and the production of several subgenomic mRNAs, the latter being translated in structural and accessory proteins [3]. New virions are assembled in the endoplasmic reticulum–Golgi intermediate compartment (ERGIC), and the virions are secreted via the secretory pathway or lysosomal exocytosis [5,6].

Entry and replication have been the two main targets for antiviral strategies against coronaviruses. The high efficacy of vaccines based on S proteins as the main antigen (either through mRNA or recombinant proteins) proved the major role played by this protein in coronavirus infections. For SARS-CoV-2 infections, the use of monoclonal neutralizing antibodies to inhibit entry has shown its efficacy in vitro and in vivo. However, their efficacies were lowered by the appearance of SARS-CoV-2 variants which were resistant to most of the commercial neutralizing antibodies [7,8]. The main targets for replication inhibitors are the viral proteases and RNA-dependent RNA polymerase (RdRp). Before being able to produce specific antiviral compounds against SARS-CoV-2, a repositioning strategy of existing compounds (antiviral or not) approved for their use in humans has been performed. Among the hundreds of compounds that have been tested, only remdesivir, a RdRp inhibitor [9], has been approved for its use in patients suffering from COVID-19. Remdesivir is administrated intravenously, which limits its use for early treatment [10]. Molnupiravir, another RdRp inhibitor, has shown interesting antiviral capacity [11]. However, its efficacy in patients is controversial [12]. More recently, Paxlovid, a combination of nirmaltrevir, a molecule inhibiting the SARS-CoV-2 protease Mpro, and ritonavir, a molecule which prevents nirmatrelvir degradation, has been launched [13]. This molecule, which was originally designed for SARS-CoV, was adapted to SARS-CoV-2. This drug has proven its efficacy by lowering the risk of progression to severe COVID-19 by 89% [14]. Many other compounds are currently being evaluated in clinical trials and hopefully will be available for COVID-19 therapy in the following years. Despite the tremendous effort of the scientific community and pharmaceutical companies to produce efficient treatments against SARS-CoV-2, few molecules are available so far. Due to the capacity of viruses to mutate and resist single-molecule therapy, it is important to identify many more antiviral compounds to be able to produce bi- or tri-therapies in the future. HCoVs are a permanent threat, and the emergence of a new highly pathogenic CoV is more than probable in the next decades.

Natural products have been shown to be a great source of antiviral molecules and hence might provide an alternative strategy for anti-CoV drug discovery [15,16]. Their major strength resides in their capacity to act on several protein targets [17]. Indeed, the antiviral activity of several natural compounds belonging to different chemical classes has already been demonstrated for CoVs, and different mechanisms of action related to structural diversity have been described [18,19,20,21]. Jo et al. highlighted the structural features required for activity using in silico analyses (hydrophobic aromatic ring, hydroxyl groups and carbohydrate moieties) [22]. For compounds without an aromatic ring, the presence of hydrophilic and lipophilic regions and the ability to form multiple hydrogen bonds through hydroxyl are needed. In addition, natural products issued from fungi or bacteria such as mycophenolic acid and sinefungin and their derivatives (remdesivir and BCX4430) have proven to be effective antivirals. Recently, many natural compounds have been also evaluated through virtual docking against SARS-CoV-2 viral proteins, but validation of their in vitro and in vivo antiviral potency will be necessary [23,24,25].

Lichen compounds are endowed with diverse biological activities, including antiviral properties, and hence can be used as potential sources of new drugs [26]. Lichens are symbiotic organisms, offering a unique chemical reservoir of depsides, depsidones, depsones, dibenzofurans or pulvinic derivatives as well as more common compounds such as xanthones and anthraquinones [27]. Furthermore, the lichen thallus is a support for other microorganisms living inside and outside the thallus, including endo- and epi-lichenic fungi and bacteria [28]. These micro-organisms could be isolated and cultured to produce other specific natural compounds. Here, we screened several classes of compounds produced by lichens or their associated bacteria and fungi against HCoV-229E and identified potent antiviral activity in four of them. One of these compounds, the depside perlatolic acid, was also shown to be active against SARS-CoV-2.

## 2. Materials and Methods

### 2.1. Chemicals

Dulbecco’s modified Eagle’s medium (DMEM) and phosphate-buffered saline (PBS) were purchased from Life Technologies (Carlsbad, CA, USA). Fetal bovine serum (FBS) was obtained from Eurobio. Remdesivir (GS-5734) and GC376 were purchased from Selleck Chemicals (Houston, TX, USA). Camostat mesylate, tariquidar, and dimethylsulfoxide (DMSO) were obtained from Sigma (Saint Louis, MO, USA). The lichen compounds were resuspended in DMSO at 100 mM.

### 2.2. Natural Lichen Products

The lichen metabolites referenced in [29] were obtained as gifts from the Huneck’ chemical library in the Berlin Museum. The other samples were previously isolated from phytochemically investigated lichens in Limoges, (Nouvelle Aquitaine, France), Rennes (Bretagne, France) or from lichen-associated micro-organisms Lichen compounds were extracted from the lichen thalli using organic solvents and purified through chromatographic processes, as described in the related references. The compounds were duly identified according to their spectroscopic features, including based on their nuclear magnetic resonance (NMR) and high-resolution mass spectrometry (HR-MS) spectra, and using polarimetry for compounds with asymmetric carbons. Microbial (bacteria or fungal) metabolites were obtained after culture of the associated bacteria [30] or the endolichenic fungi previously isolated from lichens [31], which were then purified and identified in the same way as the lichen metabolites.

### 2.3. Cells and Culture Conditions

The human hepatoma cell line Huh-7 and F1G-red, a modified African monkey kidney cell line Vero-81 with a GFP reporter gene [32], were grown in DMEM with glutaMAX-I and 10% FBS in an incubator at 37 °C with 5% CO_2_. A stable cell line of Huh-7 cells expressing TMPRSS2 was produced using a lentiviral vector expressing TMPRSS2 [33].

### 2.4. Viruses

The following viral strains were used: recombinant HCoV-229E-Luc (kindly gifted by Pr. V. Thiel) and SARS-CoV-2 (isolate SARS-CoV-2/human/FRA/Lille_Vero-81-TMPRSS2/2020, NCBI MW575140).

### 2.5. Cell Toxicity Assay

6 × 10^4^ Huh-7 cells were seeded in 96-well plates and incubated for 16 h at 37 °C. The cells were then treated with each compound, either at 25 μg/mL for the screening experiment, or at increasing concentrations for the dose–response experiment, and incubated for 24 h or 48 h at 37 °C. An MTS [3-(4,5-dimethylthiazol-2-yl)-5-(3-carboxymethoxyphenyl)-2-(4-sulfophenyl)-2H-tetrazolium]-based viability assay (Cell Titer 96 Aqueous non-radioactive cell proliferation assay, Promega) was performed as recommended by the manufacturer. The absorbance of formazan at 490 nm was detected using a plate reader (ELX 808 Bio-Tek Instruments Inc., Winooski, VT, USA).

### 2.6. HCoV-229E Infection Inhibition Assays

#### 2.6.1. Luciferase Assay

HCoV-229E-Luc was first mixed with the compounds at the appropriate concentrations. On the day before infection, 6 × 10^4^ Huh-7 cells and Huh-7/TMPRSS2 cells were seeded in 96-well plates at 37 °C. The cells were inoculated with HCoV-229E-Luc at a MOI of 0.5 in a final volume of 50 μL for 1 h at 37 °C in the presence of the different compounds, either at 25 μg/mL for the screening experiment or at increasing concentrations for the dose–response experiment. The virus was removed and replaced with culture medium containing the different compounds for 6 h at 37 °C. The cells were lysed in 20 μL of Renilla Lysis Buffer (Promega, Madison, WI, USA), and luciferase activity was quantified using a Tristar LB 941 luminometer (Berthold Technologies, Bad Wildbad, Germany) with the Renilla Luciferase Assay System (Promega) as recommended by the manufacturer.

#### 2.6.2. Time-of-Addition Assay

To determine at which step of the infectious cycle the compounds executed their effects, a time-of-addition assay was performed, for which each compound (and 50 μM of camostat mesylate, 200 nM of remdesivir and 50 μM of GC376 as a control) was added at different time points: 1 h before (referred to as the ‘pretreatment’ condition), during (referred to as the ‘inoculation’ condition) or after inoculation of HCoV-229E-Luc in Huh-7/TMPRSS2 cells. For the third condition, the compounds were added after removal of the inoculum (referred to as the ‘PI’ condition, post-inoculation), either at the end of the inoculation or 1 h after inoculation (referred to as the ‘1 h PI’ and ‘2 h PI’ conditions, respectively) and were left in the medium for the rest of the incubation time. The cells were lysed at 7 h post-inoculation, and the luciferase activity was quantified as described.

### 2.7. SARS-CoV-2 Infection Inhibition Assays

A total of 4.5 × 10^3^ F1G-red cells per well [32] were seeded in 384-well plates at 37 °C and inoculated 24 h later with SARS-CoV-2 at a MOI of 0.2 in the presence of 50 nM tariquidar and lichen compounds or GC376 at different concentrations. Images were acquired 16 h later using an InCell-6500 automated confocal microscope (Cytiva, Marlborough, MA, USA), and the percentages of infection and total cell numbers were assessed as described [32].

### 2.8. Statistical Analysis

Statistical calculations were carried out using Prism 10 statistical software (Graph Pad Inc., La Jolla, CA, USA). Non-parametric Kruskal–Wallis tests were used for statistical comparisons of groups of data. Then, individual differences between treatments were identified using Dunn’s test as a post hoc test. Prism 10 was also used for determination of the IC_50_ and CC_50_ values.

## 3. Results

### 3.1. Screening of 42 Compounds Isolated from Lichens

In order to identify new antiviral compounds against HCoVs and broaden the chemical structure diversity, 42 compounds isolated from lichens or their associated bacteria and fungi (Table 1) were tested against HCoV-229E-Luc. All the compounds were previously described in the literature (listed in Table 1) and were characterized using NMR spectroscopy and HR-MS. Huh-7 cells transduced with a lentiviral vector expressing the TMPRSS2 protease gene were used to mimic the in vivo entry pathway. In parallel, the toxicity of these compounds in Huh-7 cells was assessed using an MTS assay at 24 h. The toxicity and antiviral activity of each compound were tested at 25 μg/mL, and the results are presented in Figure 1.

As shown in Figure 1A, 6-methoxy-2-methyl-3-heptylprodiginine had a severe and significant effect on cell viability at the tested concentration (*p* = 0.021). Some of the other compounds, including enniatin B, everinic acid, perlatolic acid, psoromic acid, variolaric acid and (−)-placodiolic acid had a mild, though not significant, effect on cell viability. For the antiviral assays, five compounds were identified to strongly (approximately 2 × Log_10_) and significantly reduce HCoV-229E-Luc infection (Figure 1B): perlatolic acid (*p* = 0.020), vulpinic acid (*p* = 0.031), enniatin B (*p* = 0.022), (−)-placodiolic acid (*p* = 0.029) and 6-methoxy-2-methyl-3-heptylprodiginine (*p* = 0.011). However, as the latter two also had a greater effect on cell viability, they were not further considered for investigation in the present study. In addition, for two other compounds, chloroatranol and emodin, a reduction in infection of approximately 1 × Log_10_ was observed, although this was not significant with our test conditions. Based on these results, among the 42 tested compounds, five were selected for further investigation.

### 3.2. Determination of IC_50_ of the Active Compounds

Dose–response experiments were performed to confirm antiviral activity of the selected molecules. The toxicity of the compounds was also determined at the same concentrations at 24 h and 48 h (Figure 2A). Antiviral assays were performed for Huh-7 and Huh-7/TMPRSS2 to cover the two entry pathways (Figure 2B).

As shown in Figure 2B, the antiviral activity against HCoV-229E-Luc was confirmed for perlatolic acid, vulpinic acid, emodin, and chloroatranol, while for enniatin B, the antiviral activity could not be confirmed. The decrease in luciferase activity was weaker in the Huh-7 cells compared to the Huh-7/TMPRSS2 cells because the luciferase level was lower in Huh-7 cells than Huh-7/TMPRSS2 cells due to a reduced infection rate. However, it was confirmed that four compounds exerted an antiviral activity against HCoV-229E-Luc in Huh-7 and Huh-7/TMPRSS2 cells, demonstrating that they were able to inhibit virus infection via the two entry pathways. The CC_50_, IC_50_ and selectivity index (SI; CC_50_/IC_50_) of each compound are shown in Table 2.

The results showed that perlatolic acid and vulpinic acid were the most active, as they had the lowest IC_50_ values (16.42 ± 1.66 μM and 14.58 ± 5.55 μM, respectively). Moreover, vulpinic acid displayed a higher SI than perlatolic acid. For emodin and chloroatranol, the CC_50_ and SI values could not be calculated because the CC_50_ was higher than the tested concentrations.

### 3.3. Kinetic Study

To gain insights into the mechanism of action of the active compounds, a time-of-addition assay was performed for chloroatranol, emodin, perlatolic acid and vulpinic acid. The concentration used for this assay was fixed at approximately 2 × IC_50_. The compounds were added at different time points during infection of HCoV-229E-Luc in Huh-7/TMPRSS2 cells, either before inoculation (pre-treatment), during the inoculation (entry) or 1 h or 2 h post-inoculation (replication) (Figure 3A). Camostat mesylate, a TMPRSS2 inhibitor, was added as an entry inhibitor control. Remdesivir (RdRp inhibitor) and GC376 (M^pro^ protease inhibitor) were added as controls of the replication step. The results show that the four molecules were active when they were added at the post-inoculation step, most likely the replication step, with similar kinetics of action than remdesivir and GC376 (Figure 3B). No antiviral activity was observed when the compounds were added before or during the inoculation step, demonstrating that they are unlikely to inhibit entry. Even if the concentration used for this assay was fixed at 2 × IC_50_, the antiviral inhibition capacity was different between the molecules.

### 3.4. Anti-SARS-CoV-2 Activity

Due to their interesting effects on HCoV-229E, these molecules were further investigated for their potential pan-coronavirus activity. Therefore, the antiviral activity of chloroatranol, emodin, perlatolic acid and vulpinic acid were tested against SARS-CoV-2 in Vero-81-derived F1G-red cells [32]. The protease inhibitor GC376 was added as a control. As shown in Figure 4, chloroatranol and vulpinic acid were not active. Emodin seemed toxic at the highest concentration tested (200 μM); therefore, the antiviral activity observed was more likely due to toxicity. The only active compound against SARS-CoV-2 was perlatolic acid, which exerted a dose-dependent inhibition of infection without toxicity, with an IC_50_ value of 31.9 μM. As expected, GC376 was active against SARS-CoV-2, with an IC_50_ value of 0.22 μM.

## 4. Discussion

Of the 42 lichen-associated compounds that were tested for their activity against HCoV-229E infection in Huh-7/TMPRSS2, only four had a clear antiviral activity. However, the antiviral activity of enniatin B, isolated from an endolichenic fungus *Fusarium avenaceum* could not be validated in dose–response experiments after the first screen. This compound has already been described as cytotoxic towards several cancer cell lines [51].

Moderate activity was observed for two compounds (emodin and chloroatranol), while perlatolic and vulpinic acids displayed higher activities. Each compound belongs to a different chemical class, including depside (perlatolic acid), monoaromatic (chloratranol), anthraquinone (emodin) and pulvinic derivatives (vulpinic acid) (Figure 5). Structurally, they all consist of one or more hydrophobic aromatic ring(s) and at least one hydroxyl group, allowing hydrogen bonding with the target. The time-of-addition assay showed that these compounds inhibit HCoV-229E infection similarly to remdesivir and GC376. It is therefore likely that they inhibited the replication step, but more research will be required to identify the exact target. Replication is one of the major targets of antiviral agents used in therapy. The efficacy of nirmatrelvir, an inhibitor of the main protease of SARS-CoV-2, in COVID-19 patients is proof that HCoV infection might be successfully inhibited by these types of molecules.

Emodin, an anthraquinone compound isolated from the lichen *Nephroma laevigatum* [52], is also biosynthetized by various plants [53]. It was previously described as an inhibitor of the interaction between the SARS-CoV spike protein and the host ACE2 in a dose-dependent manner [54]. Unfortunately, we could not observe any activity of emodin against SARS-CoV-2. Our results showed that emodin is active against HCoV-229E when added after the entry step, meaning that it is most likely a replication inhibitor. For parietin, its derivative with a closely related structure, a loss of activity was noticed. The methylation of the hydroxyl in position 3 for the former may prevent the formation of the hydrogen bond. Soloronic acid might lack activity due to the presence of a methoxy group and a ketone with a pentyl chain in its structure.

Chloroatranol is an hydrolyzed compound of chloroatranorin, a depside present in tree moss (*Pseudevernia furfuracea*) and oak moss (*Evernia prunastri*) used in fragrances [29]. Due to its potent skin sensitizing toxicity, choroatranol is now prohibited from use in cosmetics by the European Commission [55]. To our knowledge, this is the first report of the potential antiviral activity of chloroatranol.

None of the dibenzofurans tested displayed any antiviral activity against HCoV-229E. The antiviral potencies of (+) or (−) usnic acids are markedly influenced by their absolute configuration. In two cases, for the inhibition of the replication of Epstein–Barr virus as well as against the influenza virus A (H1N1), the enantiomer (−) usnic acid was shown to be more active, with a higher selectivity index SI [56]. A recent paper highlights the potential of usnic acid as inhibitor against SARS-CoV-2 Mpro using a virtual screening [57]. We did not test the antiviral activity of usnic acid against SARS-CoV-2, but our results showed that it is not active against HCoV-229E. It is interesting to note that this same article revealed a significant Mpro inhibition ability of variolaric acid, which was shown herein to be inactive, at least against HCoV-229E. Moreover, a docking analysis demonstrated the ability of usnic acid and zeorin to inhibit the interaction between the spike protein RBD and ACE2 [58].

The activity against the main protease of SARS-CoV-2 of two other pulvinic acid derivatives, rhizocarpic acid and calycin, has been recently highlighted in silico among the 412 lichen compounds screened [23]. However, the in vitro evaluation conducted here on vulpinic, pulvinic and rhizocarpic acids and calycin did not confirm its potential against SARS-CoV-2 and/or HCoV-229E.

Among all the compounds investigated in this study, only perlatolic acid, a depside isolated from *Cladonia portentosa* in Ref. [42] was active against both HCoV-229E and SARS-CoV-2. This activity was lower than the cysteine protease inhibitor GC376, but synthesis of its analogues could be a good alternative to improve its activity. Perlatolic acid has already been highlighted for its anti-inflammatory activities and its ability to inhibit the prostaglandine E2 synthase-1, 5-lipoxygenase and NF-kB pathways [42,59]. A comparison with the structural features of the other four depsides tested in this study may suggested the role of the carboxylic function and the lipophilic chains in antiviral activity (Figure 6).

The esterification of erythrin and atranorin, together with the methyl group on the aromatic ring, contribute to the loss of the antiviral activity. In the same way, the lack of an alkyl chain in the depside core, such as for evernic acid, decreases the antiviral activity. Future in vitro evaluations of closely related compounds such as prasinic acid, anziaic acid, sphaerophorin or divaricatic acid will be of interest.

In conclusion, this study led to the identification of new antiviral agents against HCoVs, and one of them, perlatolic acid, was active against two coronaviruses of different genus, alphacoronavirus (HCoV-229E) and betacoronavirus (SARS-CoV-2). Interestingly, they all inhibited infection at the post-inoculation step, which was most likely the replication step. Replication is the major target of the antivirals used so far for COVID-19 treatment, with inhibitors targeting the RdRp, such as remdesivir and monulpiravir, or Mpro inhibitors, such as nirmatrelvir. Efforts should be made to better characterize the mechanisms of action of these natural compounds isolated from lichens and to identify their targets.

## Figures and Tables

**Figure 1 viruses-15-01859-f001:**
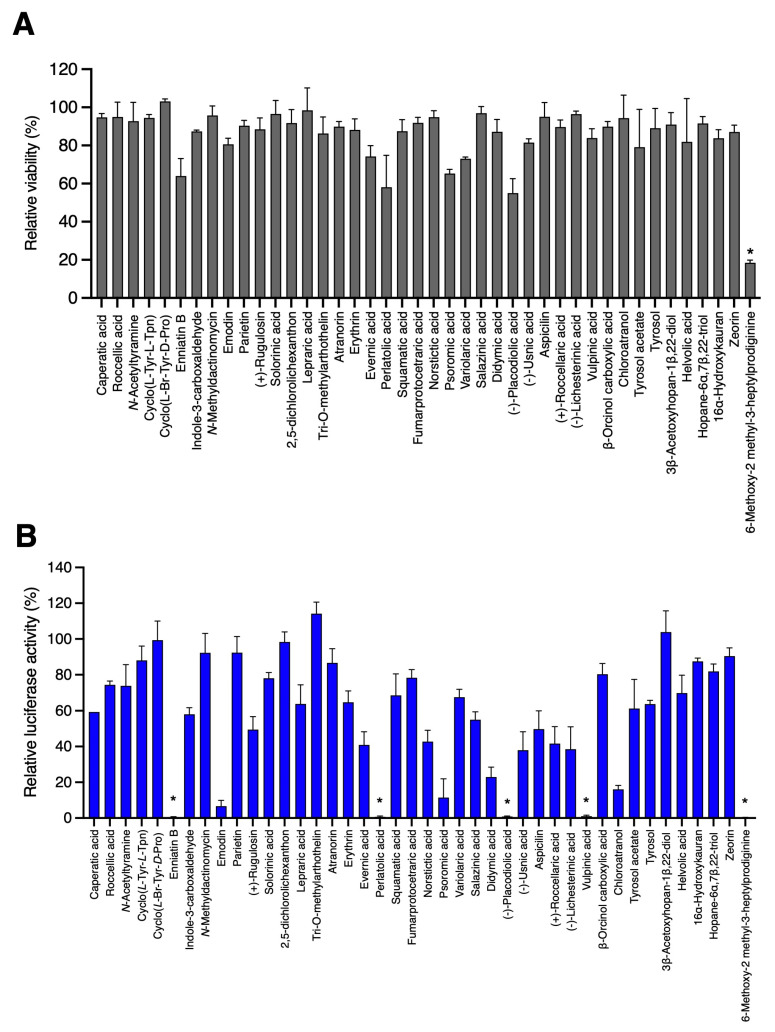
Toxicity in Huh-7 cells and antiviral activity of lichen compounds against HCoV-229E-Luc. (**A**) Huh-7 cells were incubated with each compound at 25 μg/mL for 24 h. An MTS assay was performed to monitor cell viability. (**B**) Huh-7/TMPRSS2 cells seeded in a 96-well plate were inoculated with HCoV-229E-Luc in the presence of natural compounds at 25 μg/mL. The cells were lysed 7 h post-inoculation, and luciferase activity was quantified. The data are expressed relative to the control DMSO at 0.02%, for which a value of 100 was attributed. Results are expressed as mean ± SEM of 3 experiments. Significantly different from the control (DMSO): * *p* < 0.05.

**Figure 2 viruses-15-01859-f002:**
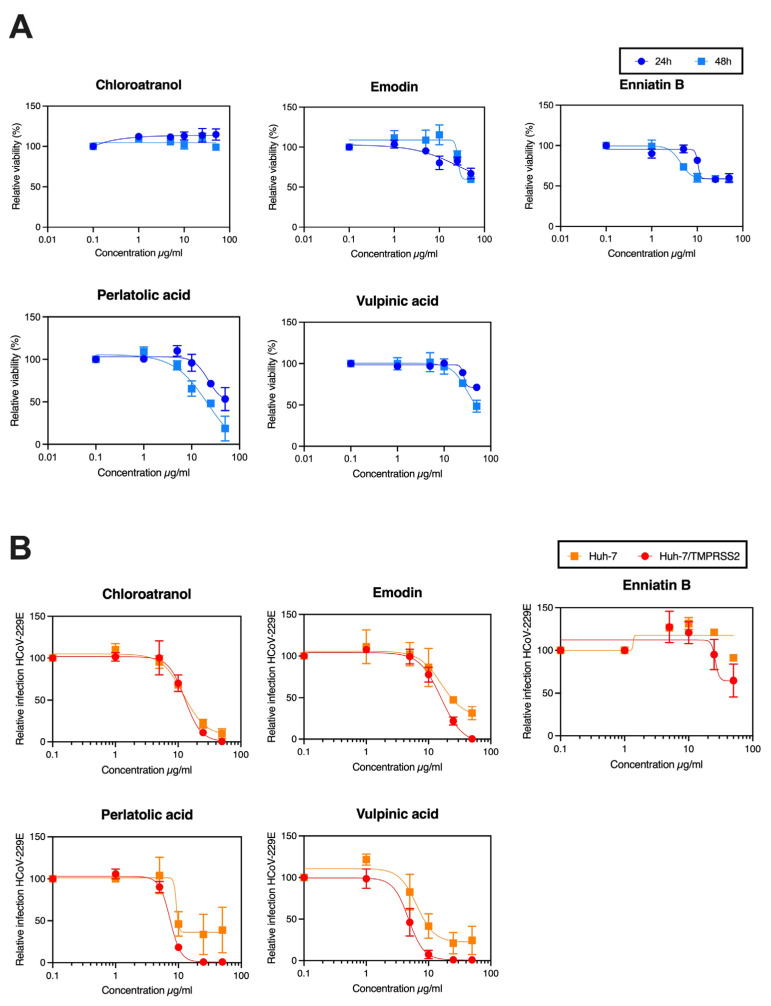
Dose–response experiments. For each compound, toxicity at 24 h and 48 h (**A**) and antiviral activity (**B**) were determined with increasing concentrations of each molecule. The method was the same as in Figure 1. The antiviral activity and the toxicity were assessed at 0, 1, 5, 10, 25 and 50 μg/mL. Results are expressed as mean ± SEM of 3 experiments.

**Figure 3 viruses-15-01859-f003:**
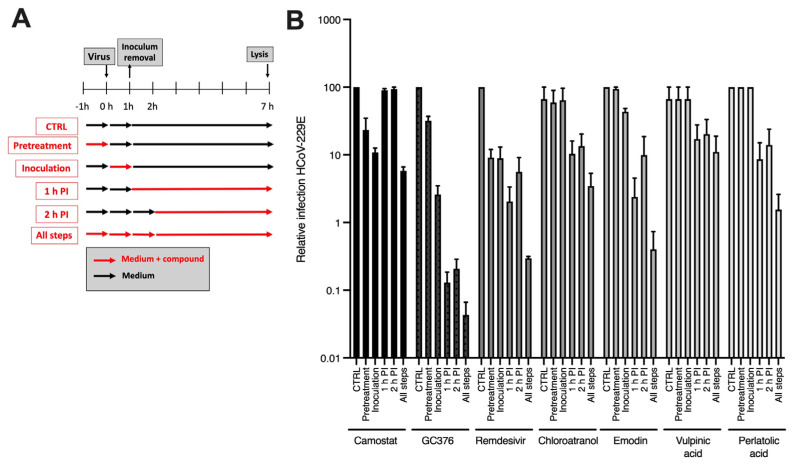
Time-of-addition assay. (**A**) Schematic representation of the experiment. (**B**) Lichen molecules (140 μM chloroatranol, 120 μM emodin, 40 μM vulpinic acid and 40 μM perlatolic acid) and controls (50 μM camostat, 50 μM GC376 and 200 nM remdesivir) were added at different time points during infection of Huh-7/TMPRSS2 cells by HCoV-229E-Luc, either 1 h before inoculation (pre-treatment), during inoculation for 1 h (inoculation), 1 h post-inoculation for 6 h (1 h PI), 2 h post-inoculation for 5 h (2 h PI), or during all the steps (All steps). Cells were lysed 7 h post-inoculation and luciferase activity was quantified. Results are the mean ± SEM of 3 experiments performed in triplicate.

**Figure 4 viruses-15-01859-f004:**
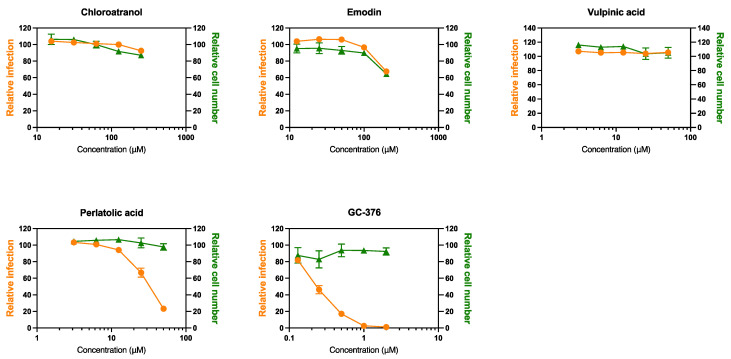
Antiviral activity against SARS-CoV-2. F1G-red cells seeded in 384-well plates were inoculated with SARS-CoV-2 at an MOI of 0.2 in the presence of 50 nM tariquidar and the compounds at the indicated concentrations. Infection was recorded 16 h post-inoculation. Images were acquired using an InCell-6500 automated confocal microscope, and the nuclei and infected cells were quantified. The graphs represent the number of infected cells and the total number of cells relative to the DMSO control of two experiments performed in triplicate.

**Figure 5 viruses-15-01859-f005:**
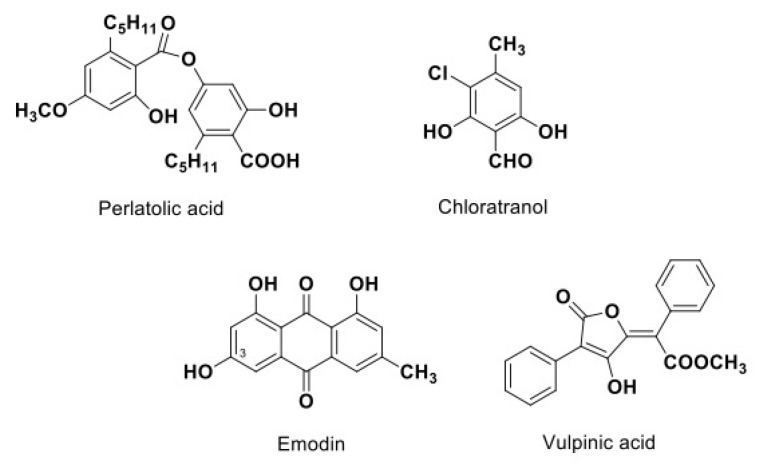
Chemical structures of the most active lichen compounds.

**Figure 6 viruses-15-01859-f006:**
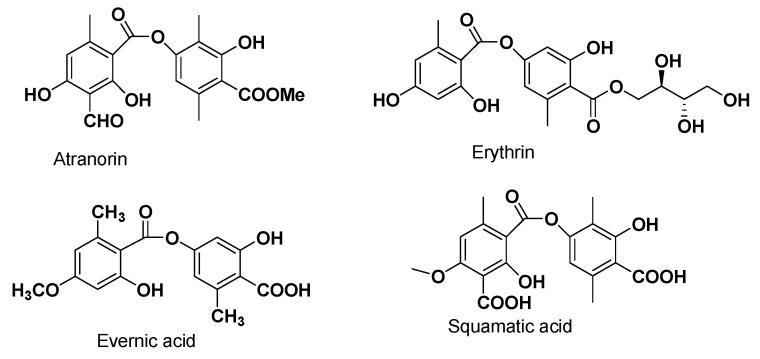
Structure of the other depsides tested in vitro.

**Table 1 viruses-15-01859-t001:** Lichen and microbial metabolites tested in vitro.

N°	Name of Compound	Class of Compound	Source	Reference
		**Aliphatic acids**		
**1**	Caperatic acid		*Flavoparmelia caperata*	[34]
**2**	Roccellic acid		*Lepraria membranacea*	[35]
		**Amino-acid** **derivatives**		
**3**	*N*-Acetyltyramine		Endolichenic fungus *Nemania aena* var *aureolatum*	*
**4**	Cyclo(*L*-Tyr-*L*-Tpn)		α-*Proteobacterium* MOLA 1416 (lichen-associated bacterium)	[36]
**5**	Cyclo(*L*-Br-Tyr-*D*-Pro)		*Nocardia ignorata* (lichen-associated bacterium)	[37]
**6**	Enniatin B		*Fusarium avenaceum* (endolichenic fungus)	*
**7**	Indole-3-carboxaldehyde		*Nocardia ignorata* (lichen-associated bacterium)	[37]
**8**	*N*-Methyldactinomycin		*Streptomyces cyaneofuscatus* (lichen-associated bacterium)	[38]
		**Anthraquinones**		
**9**	Emodin		*Nephroma laevigatum*	[39]
**10**	Parietin		*Xanthoria parietina*	[29]
**11**	(+)-Rugulosin		*Coniochaeta lignicola* (endolichenic fungi)	[40]
**12**	Solorinic acid		*Solorina crocea*	[29]
		**Chromones and Xanthones**		
**13**	2,5-dichlorolichexanthon		*Pertusaria aleianta*	[29]
**14**	Lepraric acid		*Roccella fuciformis*	[41]
**15**	Tri-*O*-methylarthothelin		*Dimelaena* cf. *australiensis*	[29]
		**Depsides**		
**16**	Atranorin		*Parmotrema tinctorum*	[29]
**17**	Erythrin		*Roccella phycopsis*	[36]
**18**	Evernic acid		*Evernia prunastri*	[29]
**19**	Perlatolic acid		*Cladonia portentosa*	[42]
**20**	Squamatic acid		*Cladonia squamosa*	*
		**Depsidones**		
**21**	Fumarprotocetraric acid		*Cetraria islandica*	[29]
**22**	Norstictic acid		*Pleurosticta acetabulum*	[43]
**23**	Psoromic acid		*Squamarina cartilaginea*	[29]
**24**	Variolaric acid		*Ochrolechia parella*	[44]
**25**	Salazinic acid		*Parmelia saxatilis*	*
		**Dibenzofurans**		
**26**	Didymic acid		*Cladonia incrassata*	[45]
**27**	(−)-Placodiolic acid		*Leprocaulum microscopicum*	[46]
**28**	(−)-Usnic acid		*Leprocaulum microscopicum*	[46]
		**Lactones and macrolides**		
**29**	Aspicilin		*Aspicilia caesiocinerea*	[29]
**30**	(+)-Roccellaric acid		*Cetraria islandica*	[47]
**31**	(−)-Lichesterinic acid		*Cetraria komarovii* Synthetic compound	[48]
		**Pulvinic acid derivative**		
**32**	Vulpinic acid		*Letharia vulpina*	[29]
		**Phenolic acid and derivatives**		
**33**	β-Orcinol carboxylic acid		*Pseudevernia furfuracea*	[29]
**34**	Chloroatranol		*Evernia prunastri*	[29]
**35**	Tyrosol acetate		Endolichenic fungi *Nemania aena* var *aureolatum*	*
**36**	Tyrosol		Endolichenic fungi *Nemania aena* var *aureolatum*	*
		**Terpenes**		
**37**	3β-Acetoxyhopan-1β,22-diol		*Pseudoparmelia texana*	[49]
**38**	Helvolic acid		Endolichenic fungi *Nemania aena var aureolatum*	[50]
**39**	16α-Hydroxykauran		*Ramalina tumidula*	[29]
**40**	Hopane-6α,7β,22-triol		*Nephroma laevigatum*	*
**41**	Zeorin		*Leprocaulum microscopicum*	[46]
		**Others derivatives**		
**42**	6-Methoxy-2-methyl-3-heptylprodiginine		α-*Proteobacterium* MOLA 1416 (lichen-associated bacterium)	[36]

* Isolated from lichens or endolichenic fungi. Millot, Mambu. Unpublished results.

**Table 2 viruses-15-01859-t002:** Toxicity, activity and selectivity index of lichen compounds against HCoV-229E.

		Huh-7		Huh-7/TMPRSS2	
	CC_50_ (μM) *	IC_50_ (μM)	SI	IC_50_ (μM)	SI
Perlatolic acid	48.9	20.81 ± 7.44	**2.3**	16.42 ± 1.66	**3.0**
Vulpinic acid	155	19.86 ± 9.46	**7.8**	14.58 ± 5.55	**10.6**
Emodin	>185	59.58 ± 1.62	**>3.1**	59.25 ± 2.47	**>3.1**
Chloroatranol	>268	65.81 ± 6.27	**>4**	68.86 ± 11.52	**>3.9**

* CC_50_ at 48 h.

## Data Availability

Not applicable.

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
