# Peer review of "Lichen or Associated Micro-Organism Compounds Are Active against Human Coronaviruses"

_viruses, 2023, doi:10.3390/v15091859_

Round 1
Reviewer 1 Report (Previous Reviewer 2)
General comment:
Thanks to the authors for the answers to the numerous questions asked, but I still have a few questions… or remarks…
Abstract
Lines 28-29: “Moreover, perlatolic acid might be further studied for its broad-spectrum antiviral activity”
I do not completely agree with this statement... Indeed, in the present study, the authors “only” demonstrated antiviral activity on Coronaviridae. De facto, I think the qualifier "broad spectrum" is a bit excessive.
Materials and Methods
Lines 139-140: “A stable cell line of Huh-7 cells expressing TMPRSS2 was produced.” Can the authors add a bibliographic reference?
Lines 158-161: “Cells were inoculated with HCoV-229E-Luc at a MOI of 0.5 in a final volume of 50 µL for 1 h at 37°C in the presence of the different compounds, either at 25 µg/ml for the screening experiment, or at increasing concentrations for the dose-response experiment.”
Lines 178-180: “4.5 103 F1G-red cells per well [32] were seeded in 384-well plates at 37°C and inoculated 24 h later with SARS-CoV-2 at a MOI of 0.2 in the presence of 50 nM tariquidar and lichen compounds or GC376 at different concentrations.”
Just a naive question: doesn't the fact of using two different MOIs induce a bias when I compare the results?
Results
Lines 245-246: “Moreover, vulpinic acid displayed the highest SI.”
Written like this, I can't completely agree with this statement…
Lines 246-247: “For emodin and chloroatranol, the CC50 could not be calculated because it was higher than the tested concentrations.” I agree with the authors. Moreover, if I look at the values in Table 2, I even wonder if the SI values for emodin and chloroatranol are not “underestimated”, since the CC50 values are outside the range of concentrations tested.
Several times, the authors affirm:
Line 251: “The concentration used for this assay was fixed at approximately 2 x IC50”.
Lines 260-262: “Even if the concentration used for this assay was fixed at 2 x IC50, antiviral inhibition capacity was different between the molecules.”
but when I look at the caption for Figure 3 it is written:
Lines 264-…: “B. Lichen molecules, 50 µM chloroatranol, 50 µM emodin, 40 µM vulpinic acid, 40 µM perlatolic acid, and controls, 50 µM camostat, 50 µM GC376, 200 nM remdesivir, were added at different time points…”
and there, I have a problem... in particular with the concentrations tested for the molecules chloroatranol and emodin... If I read correctly what is written in the legend of figure 3 and the values of table 2, I think that we are not "approximately 2* IC50"... for these two molecules...
Minor points
I think the authors should homogenize µg/ml (for example line 147) and µL (line 159)...
Author Response
Please see the attachment.

Reviewer 2 Report (Previous Reviewer 1)
the authors need to test the cytotoxicity of the 4 selected molecules on Huh7 for 48h and not 24h to assess the effect of these molecules on cell growth.
in table 2, the authors should add the CC50 values and the selectivity index.
Round 2
Reviewer 2 Report (Previous Reviewer 1)
the authors have responded to the different comments. The manuscript can be published in its present form.
This manuscript is a resubmission of an earlier submission. The following is a list of the peer review reports and author responses from that submission.
Round 1
Reviewer 1 Report
The study focuses on characterizing the antiviral activity of natural molecules against human coronaviruses. The selected molecules were screened against HCOV-229E, and the active molecules were tested for their efficacy against SARS-CoV-2.
The manuscript is well designed and scientifically explained. However, the authors should clearly mention how they isolated and characterized the natural compounds in the supplementary section. The discussion section has been conducted correctly, without over-interpretation of the results.
However, I have some major concerns :
· Cytotoxicity should be assessed after 48 h of treatment and not 24 h. Even if the infection occurs after less than 24 h, it is necessary to assess the toxicity of the molecules after 48 h of treatment in order to examine their impact on cell proliferation.
· the authors should indicate the control (vehicle) used with the % of final DMSO on the cells.
· why in figure 1B, the Y scale is logarithmic, whereas in figure 2 the scale is linear?
· the authors indicate in line 231 that the acid perlatolic is the most selective, a column on the selectivity index should be added to table 2 and the values in µg/mL should be deleted.
· The explanation of the results obtained in part 3.3 (kinetic study) is not at all clear. Why is the RLU value for the control (CTRL = infected cells non treated) 1 instead of 100?
. in line 246, the authors state that the compounds have no antiviral activity when added before or during inoculation. however, according to the results presented in figure 3, all the molecules act as input and pre-treatment agents, since the values are 1% instead of 100%. this paragraph needs to be reconsidered.
Reviewer 2 Report
General comment:
In the present manuscript by Desmarets and colleagues, the authors screened several classes of compounds produced by lichens or their associated bacteria and fungi in order to identify potent antiviral compounds active against coronaviruses.
The theme of research is very interesting, but I have some questions… or remarks…
Introduction
Lines 62-63: “New virions are assembled in the ERGIC compartment and the virions are secreted via the secretory pathway or lysosomal exocytosis”. It seems to me that the abbreviation ERGIC is not defined.
Lines 91-93: “Moreover, MERS-CoV is still circulating and emergence of a variant with higher transmissibility variant is also a threat.” I agree with the authors, MERS-CoV is still circulating, but given the dynamics of MERS-CoV infection, do the authors really think the threat is that "great"? and why?
Lines 105-107: “Recently, many natural compounds 105 have been evaluated by virtual docking against SARS-CoV-2 viral proteins too, but vali-106 dation of their in vitro and in vivo antiviral potency will be necessary.” Can the authors add a bibliographic reference?
Materials and Methods
Lines 134-136: “Human hepatoma cell line Huh-7, F1G-red, a modified African monkey kidney cell 134 lines Vero-81 with a GFP reporter gene [29] were grown in DMEM with glutaMAX-I and 135 10% FBS in an incubator at 37°C with 5% CO2”. The cellular model (i.e., Human hepatoma cell line Huh-7) is questioning for a respiratory virus…
Lines 153-155: “Huh-7 cells and Huh-7/TMPRSS2 cells were inoculated with HCoV-229E-Luc at a MOI of 0.5 in a final volume of 50 μL for 1 h at 37°C in the presence of the different compounds.” To my knowledge, for the culture of coronaviruses, it was better to have a temperature of 33°C... the in vitro infection was better... but perhaps there is an impact due to the cellular model…
I think it would be appropriate to homogenize the Materials and Methods... as an example, I think the authors should indicate for each experiment and for each cell type, the number of cells cultured (e.g. 2.6.1., 2.7…). In the same way the temperature is not specified in paragraph 2.7.
Results
Lines 187-189: “The compounds were tested at 25 μg/mL and results are presented in Figure 1 according their classes (Table 1)”. I am sorry I do not understand… why do it? And I couldn't find anything in the Materials and Methods… I only found in paragraph 2.5: “The cells were then treated with increasing concentrations of each compound and incubated for 23 h.”
Figure 1: do we have a control? For example, DMSO only? at the corresponding concentration?
Figure 2: could the authors specify the range of concentrations tested? Are we talking about toxicity or viability for the MST assay?
Table 2: How IC50/CC50 values were determined? For what reason I don't have CC50 values for Huh-7/TMPRSS2? Don't the authors believe that it would be interesting to calculate the selectivity index?
Lines 232-234: “It is important to note that psoromic acid was toxic at the active concentrations in Huh-7 cells and was not further studied.” How was the toxicity threshold defined by the authors? Why is it different for vulpinic acid?
Lines 238: “The concentration used for this assay was fixed at approximately 2 x IC50”. I believe we have a problem… If I look at Table 2, I think we have a problem for vulpinic acid if I consider IC50 and CC50 values… Aren't we in the vulpinic acid toxicity window?
Lines 246-247: “No antiviral activity was observed when compounds were added before or during the inoculation step showing that they are unlikely to inhibit entry.” Do we have an idea about the stability of the compounds in the culture medium?
Lines 323-325: “Psoromic acid is the only depsidone among the five evaluated in this study which displayed interesting activity against HCoV-229E, but due to its low SI we did not further investigate this molecule”. SI? see comments above…
Lines 311-322:
Do the authors have any hypotheses to explain these differences in results?
Lines 341-344: “Comparison with the other four depsides shows the importance of the presence of free carboxylic group in its structure, as well as the pentyl chain on the aromatic ring, in increasing its lipophilicity (Table 2; Figure 6.).” Table 2? Are you sure?